# Change Event Dataset for Discovery from Spatio-temporal Remote Sensing Imagery

**Utkarsh Mall**  **Bharath Hariharan**  **Kavita Bala**

Cornell University

{utkarshm, bharathh, kb}@cs.cornell.edu

## Abstract

Satellite imagery is increasingly available, high resolution, and temporally detailed. Changes in spatio-temporal datasets such as satellite images are particularly interesting as they reveal the many events and forces that shape our world. However, finding such interesting and meaningful change events from the vast data is challenging. In this paper, we present new datasets for such change events that include semantically meaningful events like road construction created using Sentinel-2 satellite imagery (with 10m spatial and 1 month temporal resolution). Instead of manually annotating the very large corpus of satellite images, we introduce a novel unsupervised approach that takes a large spatio-temporal dataset from satellite images and finds interesting change events. To evaluate the meaningfulness on these datasets we create 2 benchmarks namely CaiRoad and CalFire which capture the events of road construction and forest fires. The CaiRoad dataset has a total of 28015 change events with 2259 road construction events from the city of Cairo and the CalFire dataset has 2172 change events with 204 labeled fire events in California. These new benchmarks can be used to evaluate semantic retrieval/classification performance. We explore these benchmarks qualitatively and quantitatively by using several methods and show that these new datasets are indeed challenging for many existing methods. For example the best performing model has a retrieval precision@25 of 0.46 on CaiRoad benchmark.

## 1 Introduction

At the turn of the century, there were about 30 earth observation satellites orbiting the planet. Today, that number is close to 1500 and rising [6]. The massive trove of visual imagery from these satellites cannot be more timely. Be it because of rapid human development or climate change, our world is changing, and we need to understand what is changing across the planet and why.

Prior work on analyzing change in satellite imagery identifies changed pixels between two snapshots of the same location (change detection). While this is a good first step, we need to understand not just what has changed, but also why. For example, a climate scientist may be specifically interested in changes that correspond to forest fires.

Thus, instead of simply detecting changed pixels, we want to identify change events. We define a *change event* as a group of pixels over space and time that are all changed by a single event. We are interested in developing systems that can automatically detect *change events* and assign to each a semantic label that indicates the nature of the event, e.g., forest fires, road construction etc. Identifying change events is a much more challenging problem than change detection. Multiple events might be happening in the vicinity of each other, such as a road construction near a farm being harvested. Furthermore, an event may span arbitrary time periods, so pairwise image analysis may be insufficient. Detecting, characterizing and recognizing change events is thus a novel and difficult research problem.

36th Conference on Neural Information Processing Systems (NeurIPS 2022) Track on Datasets and Benchmarks.

To foster research on this problem, we need a dataset of satellite images, corresponding change events and labels for the change events. However, manually annotating change events is extremely challenging. First, change events are rare, so annotators will have to sift through hundreds of images before coming across a change event. Second, change events can span multiple months, so annotators will have to look at multiple images together. Third, annotating change events requires the annotator to segment out the pixels involved; a task that is well known to be challenging. For example, to accurately annotate changes in a $1024 \times 1024$ dimension image pair, it takes about **4.5 person-hours** [12] and we have hundred of thousands of such image pairs. As such, instead of manually annotating all change events, we propose a semi-automatic approach for annotation: we detect change events automatically, and then ask human annotators to label them with semantic categories.

We detect change events in unlabeled satellite imagery automatically using an unsupervised change detection approach. Unfortunately, existing change detection algorithms are too slow and typically designed to detect only certain kinds of changes. Our first contribution is a new *fast, unsupervised* change detection algorithm. We adapt recent advances in self-supervised learning to produce a semantically meaningful feature representation of the pixels in satellite images. Using these features, we can dramatically speed up unsupervised change detection (**48×**) while increasing accuracy (by **8%**). Furthermore, we find that these features are useful not just for detecting changed pixels but also for grouping changed pixels into events.

We run our automatic change event detection on two regions of the world: California and Cairo. We then use these automatically discovered change events to create two *event categorization benchmarks*. The goal in the first benchmark, CalFire, is to identify the set of events in California that correspond to forest fires. The goal in the second benchmark, Cairoad, is to pull out road construction events in Cairo. Both these benchmarks are created by using publicly available information about forest fires and road construction to label automatically generated change events. These labels are then verified by humans annotators. The resulting dataset and benchmarks are not just substantially larger than prior satellite image change detection datasets ([10, 11]), but also reflect a finer-grained categorization of events, going beyond mere land-use change. For example, our benchmark contains 2259 road construction events and 204 forest fire events that are hard to find in coarser land-use datasets ([27, 48]).

We use these benchmarks to evaluate several different approaches to recognize events. In particular, inspired by successes of representation learning in standard recognition problems, we evaluate different kinds of representations for spatio-temporal events. We look at both "few-shot" retrieval as well as classification tasks. Our results show that our benchmarks are challenging and require novel innovations in representation learning.

To summarize, our contributions are:

- We define the notion of *change events* and introduce the problem of detecting and categorizing change events.
- We present a novel method to automatically create a dataset of change events from spatio-temporal satellite images.
- We present two novel benchmarks on these datasets created using a semi-automatic approach with publicly available metadata and human labels for the task of semantic categorization of *change events*.

To the best of our knowledge, these are first-of-its-kind datasets and benchmarks for interesting change events in satellite imagery. Our datasets allow new insights in using computer vision to surface and analyze important change events on a planet scale.

## 2 Related Work

We first place our work in relation to existing work in remote sensing image analysis and datasets. We also look at some ideas in self-supervised learning that we use to create our datasets.

**Satellite Image Change detection.** Computer vision has always played a big role in satellite image analysis. There is a large body of work on *change detection*, i.e., the problem of detecting the changes between two temporally separated images; change detection is the first step of our pipeline as well. Classic change detection techniques are based on differences [30, 37] and ratios [28] of pixel-level

features such as raw spectral data [49], and principal components[35]. With supervision, machine learning can be brought to bear [23]. However, these classical approaches are limited by their features, which can be sensitive to irrelevant changes such as illumination.

Recent works use patch-level features rather than pixel-level features as they are more informative and more robust to pixel noise. These features can be produced efficiently using deep neural networks (DNN) trained specifically for change detection using fully convolutional networks [16], recurrent neural networks [36] or siamese convolutional networks [22], but these typically require labeled data for training. Unsupervised approaches that detect change accurately do exist, but are extremely computationally expensive.These approaches typically use techniques such as PCA [18], kernel PCA [50], clustering [52] or saliency [20], graph networks [47] along with DNNs. In this paper, we show that self-supervised features yield even more accurate results, but at a fraction of the computational cost. While self-supervision has been previously used as pretraining for supervised change detection [34], [38], ours is the first work to use it for unsupervised change detection. Our representation also outperforms other prior work on learning representations for satellite images, such as Tile2Vec [29]. The change detection masks can be used for downstream tasks such as object detection [54]. In contrast to object detection our goal is to detect/discover events that happen over a longer period of time while being unsupervised.

**Temporal Understanding of Satellite Images.** Feature representations of satellite images suffice for change detection, but not for characterizing spatio-temporal events, which require characterizing the temporal progression as well. Some existing techniques use autoencoders with 3D convolutions [32]. Others use adversarial losses to disentangle spatial context and temporal context [44]. While these methods work well for tasks such as land-cover classification, we show empirically that these representations cannot learn informative representations for change events. Additionally these methods cannot encode change events of arbitrary size (spatial) and length (time).

**Change Detection and Spatio-Temporal Datasets.** Our work is also related to change detection datasets. Many existing change detection datasets collect manual annotations for changes between paired images [12, 11, 46, 10]. Our change events instead span over longer time periods and are automatically created. Changes identified in existing datasets are often not labeled with semantic information; even when they are, the semantic information is coarse-grained [50]), whereas our change events contain finer-grained semantic information, such as road constructions or forest fires.

Our dataset is also a spatio-temporal dataset. In remote sensing and computer vision, several spatio-temporal datasets have been proposed previously. DynamicEarthNet [48] is a landcover classification dataset but also has temporal information of landcover changes. Other datasets look at specific variables over time and space such as soil moisture [15] or forest logging [19]. Our pipeline is more automatic and can be used to rapidly create such datasets at scale and at less cost.

**Self-supervised Learning for Satellite Images** Using unlabeled data to learn a representation has seen a resurgence with advances in self-supervised learning. Early work on self-supervised approaches used "pretext tasks" like solving jigsaw puzzles [40], colorization [53] or predicting rotation [21]. More recent and better performing methods such as NPID [51], PIRL [39], MOCO [25], or SimCLR [13], use contrastive learning, which learns representations by treating images and their augmentations as a single class.

Self-supervised learning is progressing rapidly in the area of remote sensing. Researchers have used a variety of information from unlabeled datasets such as location [29, 9], season [38] or texture [8], as a signal for self-supervision. With the advent of newer architectures such as transformers [17], some methods have also used them for satellite image representation learning [45]. Self-supervised learning has also been applied to remote sensing of planets other than earth [42]. In all of these cases the objects being represented are also typically full images. Here, we present self-supervised learning techniques designed for *pixels* in satellite imagery to perform change detection.

# 3   An automatic approach to detecting change events

We propose to discover semantically meaningful events from a large amount of spatio-temporal data without any supervision. Since change events are a new type of data, we first formalize their definition in Sec. 3.1. We then present an overview of our proposed pipeline to obtain change events before elaborating on the key steps (Sec. 3.2).

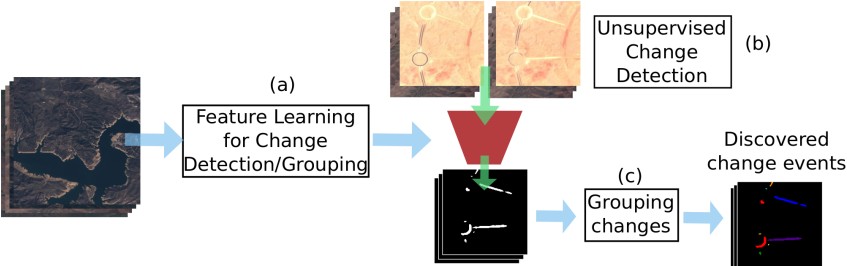

Figure 1: Overview of our method for creating the change event datasets. (a) We first use a self-supervised method to learn pixel-level features aiding in change detection and grouping. (b) We then perform unsupervised change detection for different regions and times using these features. (c) The features are also used to group these change pixels, based on similarity and proximity in space-time.

## 3.1 Formalizing Change Events

We define a *change event* as a group of pixels over space and time that were changed by a single event, such as road construction or drying up of a reservoir of water, among others. These changes can be of arbitrary shape and size in both time and space. Except for disregarding changes due to pixel noise or illumination changes (due to the changing angle of the sun), we do not place any restriction on the kind of change; this is to account for the differing needs of myriad applications. We are interested in *detecting* these change events, and also categorizing them into *classes* (e.g., road construction).

More concretely, we describe a change event using an ordered pair $\langle V_{1\cdots l}, C_{1\cdots l-1} \rangle$. Here $V \in R^{l \times x \times y \times c}$ are 3-D volumes of sequences of satellite images, where, $l, x, y$ are the span of changes in time and space and $c$ is the number of channels in satellite images. $C \in \{0, 1\}^{(l-1) \times x \times y}$ is a binary mask indicating which pixels changed between consecutive frames for the change event. $C$ has a value of $1$ whenever a change has happened otherwise it is $0$.

## 3.2 Proposed Pipeline

The input to our pipeline is a set of large spatio-temporal volumes, and the output is a set of interesting change events. Finding these events involves finding changed pixels between all pairs of consecutive images in time (Sec. 3.2.1). These changed pixels must then be grouped based on proximity in space and time and change similarity to get change events (Sec. 3.2.2). We present our pipeline (Fig. 1) to solve these challenges following these steps.

All the experiments and datasets in this dataset are created using the RGB band of Sentinel-2 satellite imagery. The spatial and temporal resolution of the collected dataset is 10m and 1 month respectively. However, our pipeline is more general than this and can be applied at any resolution or satellite images with different bands. We use these 3 bands so that we can leverage the advances in the computer vision tools that are developed on natural RGB images.

### 3.2.1 Feature Learning for Change Detection

We first learn a feature representation for pixels in an unsupervised manner. This representation can be used for both pixelwise change detection and change grouping.

Note that unsupervised change detection is not a novel problem setup. Several methods have been proposed [37, 18, 50] to perform change detection with no supervision. However, getting features that are invariant to noise and illumination is hard. Methods that work well are really slow, and thus hard to use in practice at scale. To speed up this process and maintain the feature quality on par with these methods we present a method that learns a self-supervised representation at a per-pixel-level.

This per-pixel representation should be invariant to photometric transforms or we may detect irrelevant changes such as illumination changes due to the direction of the sun. If we further want to use this representation for grouping nearby changed pixels into events, then additional invariances are required. For example, we may want to identify the construction of a road as a single event, even as it curves around (calling for robustness to rotation) or becomes wider or thinner (calling for invariance to scale changes). The construction of a sidewalk should also be considered as part of road construction (calling for invariance to translation). This range of invariances makes the problem very challenging.

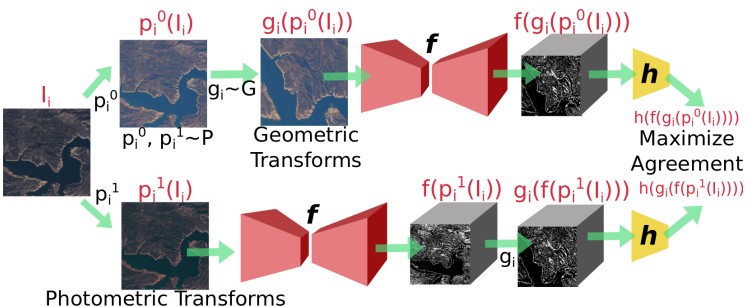

Figure 2: Training overview for our self-supervised change detection method. First, two different photometric transforms ($p_i^0$ and $p_i^1$) are applied to the image $I_i$. Then the same equivariant transform $g_i$ is applied to the top image $p_i^0(I_i)$ before feeding it into the network and also to the feature map of the bottom image $f(p_i^1(I_i))$. The agreement between the transformed feature map and the feature map of transformed images is maximized through SimCLR ($h$ is a projection head).

To learn such a feature representation, we draw on recent advances in self-supervised learning based on contrastive learning [51, 39, 25, 13]. These approaches attempt to spread images out in feature space while ensuring that augmentations (e.g., color jitter, random crops) of the same image embed close. A representative approach here is SimCLR. In each iteration SimCLR samples a batch of images and applies two augmentations to each, resulting in a combined batch $\{I_0, I_1, \cdots, I_i, \cdots, I_{2N-1}\}$, where $I_{2k}$ and $I_{2k+1}$ are two augmentations of the same image. It then optimizes the loss:

$$\sum_i \sum_{j=\{0,1\}} -\log \frac{\exp(\text{sim}(z_{2i+j}, z_{2i+|1-j|})/\tau)}{\sum_{k=0}^{2N-1} \mathbb{1}_{[i \neq k]} \exp(\text{sim}(z_{2i+j}, z_k)/\tau)} \tag{1}$$

Here sim is the cosine product, $\tau$ is a temperature, and $z_i = h(f(I_i))$ where $f(I)$ is a feature vector for image $I$ and $h$ is a projection head.

Change detection is a per-pixel classification problem. Unfortunately, SimCLR is unsuitable for our case as we need a feature representation *for each pixel* rather than a vector for the entire image. Therefore, instead of maximizing agreement between feature vectors of augmented images, we maximize agreement between feature maps. The nuance here is that photometric and geometric data augmentations are handled differently. As argued above, the *feature map* for an image should be **invariant** to photometric transforms like color jitter. When it comes to the geometric transformations mentioned above, namely, rotation, translation, and scale, we want **equivariance**, i.e., geometrically transformed feature maps should agree with feature maps of geometrically transformed images.

To obtain such invariance and equivariance, we combine contrastive learning with the augmentation framework proposed in [14]. Concretely, to learn invariance to photometric transforms, for each image $I_i$ we sample two photometric transformations $p_i^0, p_i^1$. We then produce two feature maps for $I_i$ using the two photometric transformations, $p_i^0(I_i)$ and $p_i^1(I_i)$.

To learn equivariance to geometric transform, we also sample a geometric transformation $g_i \sim \text{G}$. We first apply it on the first input image $p_i^0(I_i)$ and then obtain the feature map $f(g_i(p_i^0(I_i)))$. The same transform $g_i$ is applied on the feature map of the other image $f(p_i^1(I_i))$ to obtain $g_i(f(p_i^1(I_i)))$ (see Fig. 2). We then maximize the agreement between the geometrically transformed feature map and the feature map of the geometrically transformed image, resulting in geometric equivariance.

Both these feature maps are also passed through the SimCLR projection head $h$. This results in two feature maps $z_{2i} = h(f(g_i(p_i^0(I_i))))$ and $z_{2i+1} = h(g_i(f(p_i^1(I_i))))$. Our loss then encourages agreement between these two in a contrastive manner:

$$\sum_{j \in \{0,1\}, i, x} -\log \frac{\exp(\text{sim}(z_{2i+j}[x], z_{2i+|1-j|}[x])/\tau)}{\sum_{k \neq i, x \neq w} \exp(\text{sim}(z_{2i+j}[x], z_k[w])/\tau)}$$

Here $x, w$ index pixel locations while $i, k$ index images. The denominator includes all pixels from other images and other locations of the same image.

**Unsupervised Change Detection.** We use the learned representation $f$ to detect change over a large spatio-temporal dataset. We take pairs of temporally consecutive $m \times n$ geo-registered images,

$I_{t_1}, I_{t_2}$, and obtain a binary change map $C \in \{0,1\}^{m \times n}$ by simply thresholding the difference between the corresponding feature maps: $C = |f(I_{t_1}) - f(I_{t_2})| > \lambda$. Changed pixels have a value 1 in $C$; other pixels are 0.

This thresholding operation matches prior work on change detection[18, 50]; the key difference is in the *feature space*. In contrast to prior work, *our pretrained feature representation explicitly captures the right invariances without any human supervision* and *is fast at runtime* (as we show later).

### 3.2.2 Change Grouping

Next, we group pixel-level changes into semantically similar segments or *change events*. Note that since change events can span multiple time steps (*e.g.*, a road being constructed over months), the input to the problem is 3-dimensional. So we need to group voxels (3d-pixel) instead of pixels.

Our pre-trained feature representation allows us to use a simple off-the-shelf segmentation approach: region growing [7]. This approach starts from some seeded pixels (voxels in this case) and propagates their label to the neighboring pixels. Instead of seeding, we implement this method as finding connected components on a graph. The key here is the definition of neighborhood on the 2d pixel map of the graph. We consider two voxels $v_1$ and $v_2$ as neighbors if,

$$(d_x(v_1, v_2) + c_t d_t(v_1, v_2) < \delta_{st}) \wedge (d_f(v_1, v_2) < \delta_f) \tag{2}$$

where $d_x$, $d_t$ and $d_f$ measure distance in space, time and feature-space respectively, $c_t$ is a weighting factor, and $\delta_{st}$ and $\delta_f$ are thresholds. Thus two voxels would be grouped together only if they are both close in space and time (low $d_x$ and $d_t$) *and are semantically similar* according to our learned features (low $d_f$). Each resulting *group of changed pixels* is a change event.

## 4 Benchmarks for Change Events

We now use the above change event detection pipeline to create benchmarks for *semantically categorizing* the discovered events.

**CalFire Benchmark:** Our first benchmark looks at forest fires in California, a growing threat due to climate change that has caused substantial loss of life and property and poses innumerable health risks [31]. We create a dataset of fires using 6 years of RGB Sentinel-2 imagery (as the latest Sentinel instrument has been active since late 2015). The Sentinel-2 imagery are publicly available and free to use for any application [5]. We use the EarthEngine APIs[2], to preprocess and access the dataset.

We then obtain the location and start/end date of all forest fire incidents (1076 such locations) from the California Public Records [1]. We run our change event discovery pipeline to find 2172 change events over the 6 year span; these include fires, but also other kinds of events, such as snowfall, new construction, changes in water level, etc. The change masks for these events are further cleaned up using a CRF [33]. Next, we label each of these change events as being a fire event or not. A change event is considered a fire event if the location/time of the fire, as determined by the dataset from the public record, overlaps with the change mask of the change event, or else it is negative. We find 204 such fire events, so around 1 in every 10 change events is a fire event.

**CaiRoad Benchmark:** Our second benchmark looks at road construction events. We choose Cairo as the city in focus because it has seen a lot of new construction in the past decade. Again, we collect 6 years of Sentinel-2 images (2015-2021) from Cairo. Then, we use our pipeline to detect a wide variety of change events. Since the number of locations is much larger than CalFire locations and the amount of cloud-covered regions in Cairo are much fewer, we detect 28015 change events.

To get the ground-truth road constructions we use metadata from OpenStreetMap [3] that contains information about the history of road construction. However, this information is noisy, as the construction of some smaller roads is missing. For some roads, the construction tag is never annotated.

Therefore, we only use OpenStreetMap to check if a change event might potentially contain road construction. If the change mask event of a change event has any overlap with any road coordinates from OpenStreetMap we use these events for further filtering. This step is useful as it removes two-thirds of the change events from requiring the human annotation step. Change events such as crop growth/harvesting, building construction, changes to water level, are thus not annotated.

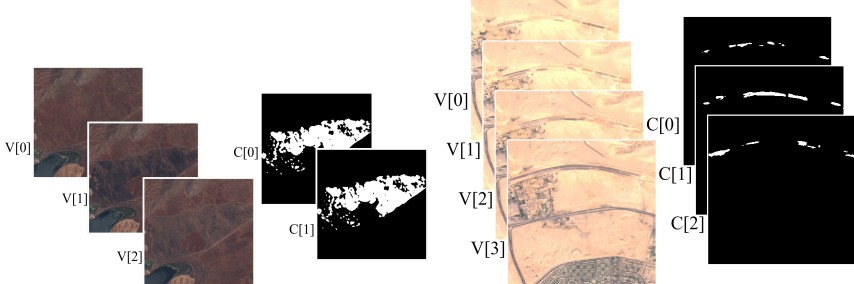

Figure 3: Example of change events corresponding to a fire (left) and road (right) in our benchmarks. The events are represented in $\langle V_{1 \cdots l}, C_{1 \cdots l-1} \rangle$ format. $V[0] \cdots V[3]$ are RGB satellite images providing visual information and $C[0] \cdots C[2]$ are binary change masks showing changes between consecutive satellite images (white is change, black is no change). For example, $C[1]$ shows the change between $V[1]$ and $V[2]$.

The remaining 9172 events are further annotated by human annotators. We use Prolific [4], to crowd-source road construction labels. Human annotators are shown change events and asked if the change event corresponds to the construction of a road or not (see supplementary for interface details). The annotators are additionally explained that construction changes can extend over multiple timesteps and are shown various examples in instructions and tutorials (see supplementary), resulting in high-quality of desirable data. On average it took about 3.45 seconds for users to understand if a change event was a road construction or not. Compensation for 100 change events was 1.85 USD. The average wage was approximately 15 USD / hour. Every change event was labeled by 3 separate annotators. An event was considered a road construction event if 2 out of 3 annotators agreed. In total these annotations cost approximately 800 USD. The annotators find 2259 such road construction events.

Fig. 3 shows examples from these benchmarks. More information about these benchmarks is in the supplementary. Note that our datasets can be used to create benchmarks for other types of change events as well. For example, we can use information from agriculture surveys (or human annotations) to label change events such as growing/harvesting of crops.

While the benchmarks are primarily aimed at representation learning for change events, it can be used for other applications too. For example, our benchmarks can be used to supervise a forest fire detection model or can be used as a dataset for supervised pairwise road change detection.

## 5 Results

In this section, we first evaluate the reasoning for using specific components in our pipeline. Then we present the methods for evaluation for using our datasets/benchmarks. Finally we evaluate change event representation learning with these metrics using existing and new methods.

### 5.1 Change detection

We first evaluate our unsupervised change detection method. We use the OSCD benchmark [11], containing satellite images around urban areas. The whole dataset (train+test split) is used as unlabeled images to train our model and baselines.

We compare our method against unsupervised change detection baselines. CVA [37] and DCVA [43] use change vector analysis on pixel and feature space respectively. PCANet [18] and KPCA-MNet [50] use a series of principal components of patches. Tile2Vec [29] learns self-supervised representations for patches. Since the range of difference can vary across methods, each method needs a different optimal threshold. Thus for fairness, we use Otsu's thresholding [41] for all the methods. The threshold maximizes interclass variance for the two classes above and below the threshold.

Note that the F-score and $\kappa$-scores are low because OSCD is annotated for urban changes, whereas our method is designed for all types of changes. The high recall score shows that our method is best at recovering the majority of changes annotated in the dataset.

| Method | F1 | Rec. | $\kappa$ | Time(s) | Method | F1 | Rec. | $\kappa$ | Time(s) |
|---|---|---|---|---|---|---|---|---|---|
| CVA [37] | 0.268 | 0.944 | 0.231 | 1.16 | DCVA [43] | 0.255 | 0.901 | 0.222 | **0.94** |
| PCANet [18] | 0.298 | 0.925 | 0.262 | 13.70 | KPCAMNet [50] | 0.302 | 0.921 | 0.264 | 54.46 |
| Tile2Vec [29] | 0.149 | 0.941 | 0.116 | 316.18 | **Ours** | **0.321** | **0.959** | **0.287** | **0.94** |

Table 1: Performance of our unsupervised change detection method on the OSCD benchmark [11]. Our method is more accurate at detecting changes (higher F-score and $\kappa$-score) and is also significantly faster than many of the better performing baselines.

Table 1 shows the results in terms of F-score and Cohen's $\kappa$-score. Our method outperforms all baselines in both accuracy and speed, and is $48\times$ faster than the closest competitor (KPCA-MNet). The speed advantage is crucial when running at scale: our approach uncovers all events from 6 years of Sentinel-2 data from Cairo in *7 hours* compared to *14 days* (est.) for KPCA-MNet (KPCA-MNet takes about 51 seconds to detect changes for a temporal pair and there are around 24k such pairs).

The poor performance of Tile2Vec shows that not all self-supervised learning techniques are useful: careful thought must be given to the needed invariances.

## 5.2 Change Event Representation Learning

We now look at how one can use our datasets and evaluate whether we can learn a representation from the proposed change events. A good representation for the change events has the potential of helping analysts and other users discover interesting events in spatio-temporal imagery. Since change events are novel objects, we evaluate a set of related existing methods and newly proposed baselines. As our goal is to learn a representation with semantic understanding of change events without labels, we focus on self-supervised methods. We look at the following baselines:

• **SimCLR: Change Event**: We learn a self-supervised representation on temporal slices of change events using SimCLR. Since we are using an architecture with global average pooling (ResNet), we replace the global average pooling layer with a weighted average pooling. The weights for a temporal slice $V_k$ of a change event $\langle V_{1...l}, C_{1...l-1}\rangle$ is its change mask from previous and next slice *i.e.* $C_{k-1} \wedge C_k$. This allows the feature to only encode regions where change is happening.

• **3DConv-AE: Change Event** [32]: It learns an unsupervised representation for satellite imagery using a reconstruction loss using an autoencoder with 3D convolutions encoding temporal information.

• **Tile2Vec**: It is an unsupervised representation learning method for satellite images with triplet loss.

• **Pretrained ImageNet**: It is a network pretrained on ImageNet. No training is involved.

• **SimCLR: EuroSat**: Instead of using temporal slices from change events we instead use images from EuroSat [27] land-cover classification. The resolutions of the images is same and the images contain various types of classes such as "forest", "highways" etc.

• **SimCLR: All Data**: It uses images from regions from which we obtain change events instead of the change events themselves. Note that this dataset is significantly larger than the change event dataset, and contains all the information that the change event dataset has.

For all the SimCLR methods and Pretrained ImageNet, we use Resnet-18 [26]. We also finetune a pre-trained ImageNet model rather that training from scratch as that helps the training. For Tile2Vec and 3DConv-AE we use the same architecture as in the those works.

Note that all the methods except for 3DConv-AE cannot encode temporal information. So to encode temporal information we first average the features across time for all these methods. While averaging the features across time will flatten the time dimension, temporal information is not completely lost since the change pairwise masks are explicitly provided to the network. We also look at alternative approaches for temporal feature aggregation in the supplementary material. More details about the model architecture are presented in the supplementary.

**Evaluation Protocol:** We evaluate our method in two ways. (a) We use a few change events of a type as a query to retrieve other events of that type using a nearest neighbor classifier on the learned representation to measure the retrieval performance. (b) We train a linear classifier on the aggregated features using a training split and measure the classification accuracy on the unseen split.

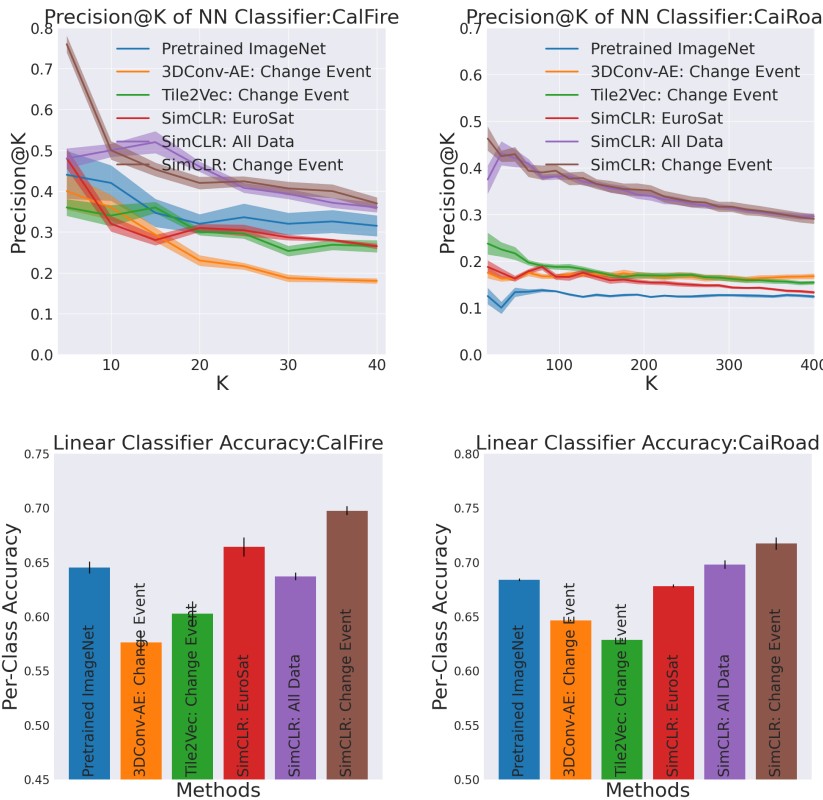

Figure 4: **Top:** Retrieval Performance on CalFire and CaiRoad benchmarks. Precision@K for the kNN classifier (k=10 for kNN). **Bottom:** Linear classifier accuracies on various self-supervised representations (except for ImageNet) on the CalFire and CaiRoad Benchmarks.

## 5.3 Retrieval and Classification Results

Fig. 4 (Left) shows the performance of retrieval of similar events using our representations on CaiRoad and CalFire. For both these datasets we randomly select k=10 query samples (10 road construction events or 10 fire events) and perform retrieval on the remainder of the dataset. Precision@K measures precision of fires or roads retrieved when top-K events are retrieved. The error-bars in the plot shows the variance with 5 different randomly initialized model (not for ImageNet). Additionally, we also randomly select query samples multiple times to measure variance. Note that even though query samples are randomly selected, they are the same for all methods.

*SimCLR:Change Events* on the change events performs the best for the retrieval task when compared to other self-supervised methods such as Tile2Vec or 3DConv-AE. Also using the informative change event dataset is more useful than using other similar informative data such as EuroSat. Finally, using *SimCLR:Change Events* performs similar to *SimCLR:All Data*. This shows the informative nature of the change events, as even with a very small subset of the full dataset, the performances are similar. The gap between *SimCLR: Change Event*, *SimCLR: All Data* and other baselines is larger for CaiRoad. We believe this is because road retrieval is a relatively easier task than forest fire retrieval if the representations are good. Forest fires in different areas and phases look very different and are thus harder to represent without supervision.

Fig. 4 (Right) shows the performance of learning a linear classifier for fire and road event classification on the CalFire and CaiRoad benchmarks. We split the full datasets in a 50-50 train-test split (available with the benchmark). We train a linear classifier on the training set and measure per-class accuracy.

Again, *SimCLR:Change Events* perform the best. On CalFire, *SimCLR: EuroSat* performs better than *SimCLR: All Data* and this is not the case for CaiRoad. This may be because the domain difference between EuroSat and CalFire is smaller than that between EuroSat and CaiRoad. CaiRoad has a lot

of satellite images of urban scenes and a drier climate as opposed to EuroSat or CalFire. This results in a similar performance when provided supervision. Tile2Vec and 3DConv-AE cannot produce very good representations.

## 6    Discussion

**Societal Impact:** A potential concern with analyzing satellite images is privacy. We intentionally use satellite images of low spatial and temporal resolution so as to not violate individual privacy. However, the road construction dataset might discover roads in private properties. Our change detection, as other change detection approaches, can potentially be used on high-resolution imagery to surveil individuals and private property. Such uses should be regulated and/or discouraged.

**Limitations.** One limitation of focusing on change events is that they are small in number, which is a problem for large models (e.g., Masked Auto-Encoders [24] ). Future work could scale up the event detection to the country or planet scale, resulting in a larger dataset and more effective training. The change detection/grouping currently requires careful hyperparameter selection and in the future more analysis is required for automated selection. Our datasets are not labelled exhaustively for all types of possible change events as exhaustive labelling is a significantly expensive process. In the future we would like to look at an efficient and exhaustive labelling of change events.

## 7    Conclusions

We define change events as a group of pixels in space-time that are changed by a single real-world event. We present two datasets of change events in satellite images, namely CaiRoad (with 28015 change events) and CalFire (with 2172 change events). We present a framework to automatically discover such spatio-temporal events from large-scale satellite images without any supervision. While self-supervised representation learning methods for change events learn a good representation for change event understanding (CaiRoad: precision@25 is 0.46 and CalFire: precision@10 is 0.50), more work in the future is required to better encode these change events.

**Acknowledgements.**    This research is based upon work supported in part by the Office of the Director of National Intelligence (Intelligence Advanced Research Projects Activity) via 2021-20111000006 and NSF 1900783. The views and conclusions contained herein are those of the authors and should not be interpreted as necessarily representing the official policies, either expressed or implied, of ODNI, IARPA, or the U S Government.

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
