# OpenReview forum: "Change Event Dataset for Discovery from Spatio-temporal Remote Sensing Imagery"
_NeurIPS.cc/2022/Track/Datasets_and_Benchmarks — NeurIPS 2022 Datasets and Benchmarks _

### Official Review · Reviewer_h7QK · 2022-07-20
**A strong contribution, providing a generative approach do semi-automatically curate novel change detection datasets**

**Rating:** 8
**Confidence:** 5
**Clarity:** The paper is well written and clear.

**Strengths:**

### Significance:
* The work is introducing, formalizing and addressing the interesting new concept of “change events”, rather than solely providing incremental improvements over existing tasks or datasets.
* The work proposes a pipeline of generating novel datasets, rather than solely releasing new data. This is promising, as it allows the audience to curate their own datasets based upon the proposed approach. The authors demonstrate their pipeline by collecting and releasing two small Earth observation datasets for change detection themselves.
* To my best understanding, the proposed approach should be sufficiently generic and domain agnostic, such that it may extend beyond satellite imagery and also be applicable to e.g. co-registered repeated measures in biomedical imaging and other fields.

### Relevance:
* The presented generative approach eases the burden of manually labeling data. (Semi-) automating the annotation process in creating datasets, in particular with respect to pixel-wise annotations, is much welcome.
* Individual components of the overall pipeline are properly benchmarked, outperform existing baselines in terms of performance and scalability. The latter is important for any exploratory (semi-)automatic data curation process in order to traverse through large data bases.

###  Accessibility and Accountability:
The code will be released in a public git repository (not released at the time of this review), the two curated datasets are hosted on a university server for public access. For the proposed pipeline to prove beneficial to the community, it will be critical the released code (underlying the proposed dataset generation process, the submission’s main contribution) is be user-friendly and allowing for a flexible usage on diverse raw data.

###  Ethical and social implications:
The authors discussed potential concerns of eventually detrimental societal impact. I do agree with the authors’ assessment that their choice of medium-resolution satellite imagery over the selected areas minimizes potential violations of citizen privacy and do have no further concerns beyond this matter.

**Weaknesses:**

###  Significance:
* The notion of “change events” and their self-supervised identification is primarily of interest during the process of dataset creation. Once dataset curation is accomplished, the inferred change maps are of their assigned semantics and can, in terms of their structure and ideally annotation quality, hardly be differentiated from conventionally obtained and manually annotated change maps.
* The two provided change detection datasets are mono-class. In theory and to the best of my understanding, the pipeline should be applicable to multi-class change events (i.e. mutiple event types within a single area). Fig. 4 in the supplementary material highlights a sample of different change type segments. However, no further experiments on multi-class change event detection are conducted in the submission, so I am not too certain about its effectiveness. Further comments by the authors on this matter may be insightful and welcome.

### Relevance:
* The two released datasets, curated for the sake of demonstrating the proposed change event discovery method, are very specific in terms of their locations rather than providing global coverage. While this may raise doubts about the representativeness and generalizibility of benchmarked models, it is of limited concern to me as the submission’s main contribution is in the dataset generator method.
* To my understanding, the proposed approach should be sufficiently generic and domain agnostic, such that it may extend beyond satellite imagery and also be applicable to e.g. co-registered repeated measures in biomedical imaging and other fields. While I consider the submission a valuable contribution for the Earth observation community, to go beyond and extend to other domains may have further increased the submission’s relevance. I would appreciate the authors comments about the feasibility of this, and potential limitations or obstacles when considering other domains.

### Accessibility and Accountability:
* My main concern is about the feasibility of hyperparameter tuning during change grouping, as this poses a risk to the general applicability of the proposed method to diverse and heterogeneous scenes or events. The authors are very honest and forthcoming about this limitation tho and acknowledge it in the main text. However, I’ll raise some questions that I consider important for the audience to assess the methods’ feasibility on a future user’s data of interest:

    a) How did you explore and choose the hyperparameters in equation (2)? Specifically, how were values such as the 20 voxels spatial thresholding, as denoted in section 3.2 of the supplementary material, chosen?

    b) How easy or sensitive are the hyperparameters to tune? The submission proposes a single hyperparametrization applied onto both sample datasets. But to which extent did you encounter different optima in between your two proposed datasets?

    c) How is the seed pixel, as mentioned in [p.6, l.198] of the main text, set?

* The image data are provided as jpg / png. This is a rather unusual format for satellite imagery, which is geo-referenced and typically provided in raster format (e.g. [*.(geo)tiff](http://docs.opengeospatial.org/is/19-008r4/19-008r4.html)). This somewhat goes against conventions, making the provided datasets harder to handle by conventional and established manners. However, the authors do provide meta information (pertaining to acquisition dates and area observed), see [here](https://www.cs.cornell.edu/projects/satellite-change-events/static/data/Readme.txt). Nonetheless, I propose addressing the above issue by also releasing the datasets in raster format.

* It is surprising that Sentinel-2’s SWIR band is not utilized in generating the CalFire benchmark, given that infra-red bands are particularly beneficial detecting thermal activities. This raises some questions:  Is there any reason your proposed pipeline may not be applied on the entire spectrum of data? Did you attempt running your pipeline on all Sentinel-2 bands, and if so, did you observe any material difference in performances? Did the benchmarked baselines have access to the full multi-spectral information, and if not, is there any reason to suspect running them solely on the RGB data may put them at a disadvantage? What’s the differences in empirically measured computational run times, when including all bands?

### Ethical and social implications:
None.

**Additional Feedback:**

What follows is mostly questions (sorted descendingly by importance), who might be of potential interest for the audience:

* How would the two closely related (but contrary) change events of “crop growth” and “crop harvesting” (example from the supplementary material) be handled? Or alternatively, the change events of “receding water level” (example from the supplementary material) and “increasing water level”? Would each pair be summarized into a single change event cluster? If so, how could automatic clustering differentiate between their opposing directedness? Is there any potential way to account for such kinds of causality?

* It would be interesting to run the proposed pipeline on recently published change detection datasets (both released after or close to submission deadline) such as [1, 2], in order to compare change map agreements and evaluate how close the proposed method comes to human annotations.

* Section 2.4 "Preprocessing/cleaning/labeling" in the supplementary material specifies that cloud-covered pixels are filtered. How is smoke in the curated forest fire dataset handled?

[i] Cha, M., Huang, K. W., Schmidt, M., Angelides, G., Hamilton, M., Goldberg, S., ... & Piou, J. (2022). MultiEarth 2022--Multimodal Learning for Earth and Environment Workshop and Challenge. arXiv preprint arXiv:2204.07649. https://sites.google.com/view/rainforest-challenge

[ii] Toker, A., Kondmann, L., Weber, M., Eisenberger, M., Camero, A., Hu, J., ... & Leal-Taixé, L. (2022). DynamicEarthNet: Daily Multi-Spectral Satellite Dataset for Semantic Change Segmentation. In Proceedings of the IEEE/CVF Conference on Computer Vision and Pattern Recognition (pp. 21158-21167).

**Correctness:**

Yes to all of the above assessment criteria. But please check to always make explicit that the dataset generation method is semi-automatic. This is not handled entirely consistently throughout the main text, see:

	“[...] we propose a semi-automatic approach: we detect change events automatically, and then ask human annotators to label them with semantic categories.” [p.2, l.40-21]


	“We present a novel method to automatically create a dataset of change events from spatio-temporal satellite images.” [p.2, l.68-69]

**Documentation:**

Data collection and organization, availability and maintenance are sufficiently detailed in the supplementary material. I validated that data is accessible and found its organization as described in the documentation. Licensing and maintenance plans are specified as well in the accompanying datasheet.

**Ethics:**

None.

**Relation To Prior Work:**

Relevant publications are cited. Closest related is probably prior work on open-set self-supervised change detection, as in [35, 40]. Related methods are referenced, compared and benchmarked against their corresponding counterparts in the proposed pipeline.

**Summary And Contributions:**

The submission introduces the **novel concept of “change events”** in satellite images, together with a **scaleable (semi-)automatic pipeline to curate future change detection datasets**. To illustrate the feasibility of their approach, the authors curate **two new change detection datasets** (California wildfire & Cairo road construction). Besides, the developed pipeline makes use of some minor but handy methodological tricks, like learning photometrically invariant & geometrically equivariant representations to deal with domain- specific noise and applying SimCLR to feature maps. Key contributions are highlighted in **bold** font.

I judge this to be a strong submission and recommend for publication. The work convinces with a sound and interesting concept, supported by solid experiments. My review highlights a few points that would be beneficial to have discussed and raise awareness of, but they are relatively minor concerns that do not overly affect my positive view of the submission. I am familiar with the subject, and the conducted experiments as well as the provided results support the certainty of my assessment.

---

> ### Author Response · Authors · 2022-08-12
> **Review Response**
>
> Thank you for your constructive feedback. We address your concerns below.
>
> ## Weaknesses
> ### Significance
> **Difference from Change maps**: A change event differs from conventional change masks in many ways. Unlike change masks, it can span multiple timesteps. A change event has semantics as it happens due to a real-world event (fire). Change events could theoretically be  created using manually annotated change maps but would require much more human expense. A human would have to annotate change masks and the semantics over multiple timesteps. In our pipeline, these steps are automatic.
>
> **Binary labels**: Exhaustive labeling of all the change events would be expensive and time-consuming, but we agree that it is a limitation of our dataset discussed in the limitation section). Researchers interested in other kinds of change events can simply use the change events from our pipeline and label them with their own labels.
>
> **Representativeness/generalizability**: Because the two datasets are from very different regions (continents, weather patterns, and cultures), we believe that the model can generalize well. But we agree that the dataset generator method can be used in the future to create a more representative dataset.
>
>
> **Change event in other domains**: Finding interesting events from a large corpus of spatio-temporal data is an interesting problem in general. We have preliminary results on video datasets such as “fingerspelling” or “cooking” where we find a useful part of a fingerspelling clip focussed on fingers, or  atomic actions such as chopping in cooking videos. Unfortunately, we could not find a big enough dataset for biomedical images to apply to our pipeline, but we would be happy to add results on video datasets if needed.
>
> ### Accessibility and Accountability
>
> **Hyperparameters**:
>
> b. **Hyperparameter Sensitivity**: The benchmark is not sensitive to hyperparameters within a reasonable range ($\delta_{st}=3, 30$, $c_t=10, 2.5$). Reducing $\delta_{st}$ to 60m leads to slightly more events 2203 (from 2172). The small 1.4% increase is because region growing is not sensitive to the hyperparameters. Many components that were connected with  $\delta=20$ still remain collected with $\delta=3$. Additionally, experiments on these datasets with different hyperparameters do not significantly change performance. We have added this information.
>
> a. **Hyperparameter Choice**: The hyperparameter values were selected by qualitatively inspecting created events on a subset of input (2 temporal tiles). We have added this information to the revision. The dataset is not sensitive to the values (see above).
>
> c. **Seeded Region Growing**: As stated in the supplementary we use the connected components method for region growing, but the two methods are equivalent in terms of the results. Also, note that the final segmentation results of seeded region growing are the same irrespective of the random seed we choose (see [7]).
>
> **Image Format**: Typical image formats makes for  ease of visualization and using standard vision tools. However, we will release the dataset in geotiff format also.
>
> **Using other bands**: We agree that using the SWIR band could be useful for fire detection. But the advantages to using the RGB band are (1) many existing vision tools such as architectures, pre-trained weights, etc. Using other bands would require retraining all the models from scratch leading to poor representations. (2) Using other bands for annotations would require experts interpreting and would be more expensive (in CaiRoad humans looked at the RGB images). (3) our dataset does not limit other researchers from using our annotations and creating a dataset with more bands. We believe our work is complementary. Fairly evaluating on a benchmark with more bands might be out of the scope of this work, but we will be happy to release the dataset with all the bands.
>
> ## Correctness
> The semi-automatic approach is for labeling the dataset, however, the unlabelled dataset is created with an unsupervised and automatic approach. So both these statements are correct. We have clarified this.
>
> ## Additional Feedback
>
> **Reversed change events**: Two Change events that are exactly reverse of each other are going to have the same representation with mean aggregation. The best performing representation with mean is not aware of the direction and we need better temporal aggregation strategies. For example, a temporally aware aggregation strategy can be the mean of the concatenation of consecutive slices. Another example could be using DTW.
>
> **Pipeline on DynamicEarthNet**: We plan to run our approach on these datasets, and we will update this post once we have the results.
>
> **Smoky change events**: In all cases, the positive fire change events represent the before and after images of the fire and not during the fire. So the smoke pixels are not visible in the event itself.
>
> New revisions in the paper are colored blue during the discussion period.

---

> > ### Comment · Reviewer_h7QK · 2022-08-20
> > **Reviewer Reply**
> >
> > This is in response to the authors' reply and to comment on selected points:
> >
> > * * *
> >
> > > **Change event in other domains**: Finding interesting events from a large corpus of spatio-temporal data is an interesting problem in general. We have preliminary results on video datasets such as “fingerspelling” or “cooking” where we find a useful part of a fingerspelling clip focussed on fingers, or atomic actions such as chopping in cooking videos. Unfortunately, we could not find a big enough dataset for biomedical images to apply to our pipeline, but we would be happy to add results on video datasets if needed.
> >
> > I do think that sharing preliminary results in other domains might be beneficial 1) in illustrating the proposed pipeline's applications beyond the exemplified use cases in Earth observation and 2) showing that the method is reasonable robust in hyperparameter selection even across domains (also see the point on *Hyperparameter Sensitivity* below). In that sense, sharing such preliminary outcomes would help making a point in both cases.
> >
> > With respect to biomedical data, while I don’t require it to be addressed in the context of this rebuttal (it’s rather a work on its own), the following data of repeated measures may be of interest:
> > - the upcoming repeated measures of [UK Biobank](https://www.nature.com/articles/s41467-020-15948-9) and, to a limited extent, …
> > - other longitudinal imaging studies such as [OASIS-3](http://www.oasis-brains.org)
> > - or task/resting-state functional MRI time series such as [1000 Functional Connectomes Project](http://fcon_1000.projects.nitrc.org/fcpClassic/FcpTable.html)
> >
> > > b. **Hyperparameter Sensitivity**: The benchmark is not sensitive to hyperparameters within a reasonable range ( δ s t = 3 , 30 ,  c t = 10 , 2.5 ). Reducing  δ s t  to 60m leads to slightly more events 2203 (from 2172). The small 1.4% increase is because region growing is not sensitive to the hyperparameters. Many components that were connected with  δ = 20  still remain collected with  δ = 3 . Additionally, experiments on these datasets with different hyperparameters do not significantly change performance. We have added this information.
> >
> >
> > I appreciate the added information, as it addresses my concern about the proposed approach potentially being applicable to novel data only involving (overly tedious) finetuning—which would have diminished the method‘s benefits over manually labeling, to some extent. So I am glad this is not an issue.
> >
> >
> > > **Image Format**: Typical image formats makes for ease of visualization and using standard vision tools. However, we will release the dataset in geotiff format also.
> >
> > > **Using other bands**: We agree that using the SWIR band could be useful for fire detection. But the advantages to using the RGB band are (1) many existing vision tools such as architectures, pre-trained weights, etc. Using other bands would require retraining all the models from scratch leading to poor representations. (2) Using other bands for annotations would require experts interpreting and would be more expensive (in CaiRoad humans looked at the RGB images). (3) our dataset does not limit other researchers from using our annotations and creating a dataset with more bands. We believe our work is complementary. Fairly evaluating on a benchmark with more bands might be out of the scope of this work, but we will be happy to release the dataset with all the bands.
> >
> >
> > I thank the authors for their efforts, and wish to underline that releasing the curated data in the current format plus in geotiff including multi-spectral bands would be strongly desirable. This way, the datasets might be available in a format approachable for established computer vision methods and also be attractive for more specialized Earth observation research, in order to maximize the audience for the curated data.
> >
> > * * *
> >
> > In sum, the authors’ reply affirms my verdict and I recommend for acceptance, this is a strong submission. I appreciate the authors having addressed my concern that the proposed pipeline is reasonably insensitive to hyperparameter selection. This furthermore indicates the proposed pipeline be applicable to diverse use cases beyond what’s evidenced in the submission, as stated in paragraph *“Change event in other domains”* of the authors' comment. The proposed semi-automated labeling pipeline is the main contribution, and it may assist a broader audience curating their own future datasets by easing pixel-wise labeling efforts. Part of what's addressed in section *"Additional Feedback"* points to potentially interesting future research.

---

### Official Review · Reviewer_SgiD · 2022-07-22
**Interesting change event benchmarks and datasets**

**Rating:** 7
**Confidence:** 3
**Correctness:** Yes.
**Clarity:** Yes.

**Strengths:**

1. The concept of "change event" is an interesting new idea in remote sensing change detection.

2. The change event detection pipeline (including the unsupervised change detection method) is clearly described.

3. The paper and dataset documentation are well written.

**Weaknesses:**

1. The proposed self-supervised learning based unsupervised change detection method is not very novel.
2. The proposed datasets lack confidence analysis about both the event type label and the event pixel coverage. The event type can be rather accurate with human-like annotation, but the event pixels I am not sure. The results on OSCD dataset show an F1 score of only 0.3, which is not very good.
3. The code is an important component of this paper as it proposes a dataset generation pipeline, but it is still unavailable.

**Additional Feedback:**

I like the idea of grouping changing pixels into change events and the paper is very well written. However, If seeing it as a dataset or benchmark paper, only the event type is (able to be) benchmarked. The other very important component from my point of view, the changing pixels, are not confident and can not be evaluated as there's no ground truth label. My evaluation of this paper is not very certain, and I look forward to author responses about my concerns.


**Documentation:**

Yes.

**Ethics:**

No.

**Relation To Prior Work:**

Yes.

**Summary And Contributions:**

The paper introduces the concept of "change event" and presents an unsupervised method for change event detection. Two benchmarks for road construction and forest fires are created and evaluated.

---

> ### Author Response · Authors · 2022-08-11
> **Review Response**
>
> Thank you for your constructive feedback. We address your concerns in the following response:
>
> ### Weaknesses
> 1. The loss function itself (SimCLR) is not a novel contribution of the paper. But the architecture for change detection and the way that SimCLR is used at individual pixels, efficiently using equivariant geometric transforms, is a novel contribution of this work.
>
> Additionally, the novel method for self-supervised change detection is not the only contribution of the paper. We also have following contributions (1) We introduce the notion of change events, (2) We present a novel method to automatically create a dataset of change events, and (3) We present 2 novel benchmarks on these datasets (CaiRoad and CalFire),
>
> 2. The reason for the low F1 score on the OSCD dataset is that the change labels are not exhaustive of all types of changes, as only urban changes are annotated. Our method’s high recall shows that it recovers a big fraction of the annotated changes while having a higher F1 score than the baselines (Please refer to sec. 5 and fig. 4 for such examples in the OSCD dataset.).
>
> There is no fair way to use existing change detection benchmarks to measure the confidence of pixel maps as all these benchmarks only label specific types of changes. In order to evaluate the confidence over event pixels, we plan to do an experiment with humans. The idea would be to ask humans to compare event maps from our method and a baseline. We will update the results once the framework is ready and the experiments are conducted.
>
>
> 3. The code is now available and can be found at (https://github.com/utkarshmall13/satellite-change-events). It contains methods to obtain the Sentinel-2 dataset, the pipeline to create our change event dataset, and the baselines we used on our benchmark.
>
>
> New revisions in the paper and supplementary are colored blue during the discussion period.

---

> > ### Comment · Reviewer_SgiD · 2022-08-22
> > **Reviewer reply**
> >
> > ### 1.
> > > The loss function itself (SimCLR) is not a novel contribution of the paper. But the architecture for change detection and the way that SimCLR is used at individual pixels, efficiently using equivariant geometric transforms, is a novel contribution of this work.
> >
> > It can be true to use this specific combination, but implementing InfoNCE loss at pixel level and equivariant geometric transforms, either is not the first attempt in Earth observation. However, since the contributions are not limited to the method, this is not a big issue. But for the authors' interest: there do exist self-supervised change detection works like [1,2]. Therefore, the sentence on Page 3 line 100 (the revised version):
> >
> > > While self-supervision has been previously used as pretraining for supervised change detection [34], [38], ours is the first work to use it for unsupervised change detection.
> >
> > Should be modified.
> >
> > ### 2.
> > > The reason for the low F1 score on the OSCD dataset is that the change labels are not exhaustive of all types of changes, as only urban changes are annotated.
> >
> > This could be a good reason. Though I am not 100% confident about the evaluation results, the supplementary result (Fig 4) gives solid support.
> >
> > ### 3.
> >
> > > There is no fair way to use existing change detection benchmarks to measure the confidence of pixel maps as all these benchmarks only label specific types of changes. In order to evaluate the confidence over event pixels, we plan to do an experiment with humans.
> >
> > I agree that the only possible way to evaluate on pixel level is human operation. This is much more work but would be very beneficial to support the proposed two datasets. The current change mask looks like two "products" instead of solid datasets that can be confidently used by further researchers who are interested in pixels.
> >
> > ### 4.
> > Glad to see codes are now available.
> >
> > ### Summary
> > In general, the authors addressed most of my concerns and promised further work on pixel-level confidence. The main contribution from my point of view is the proposed change event generation pipeline, of which the code has been available (and from the revision it should generalize well). Therefore, I would like to increase my score and recommend acceptance.
> >
> > [1] Saha, Sudipan, Patrick Ebel, and Xiao Xiang Zhu. "Self-supervised multisensor change detection." IEEE Transactions on Geoscience and Remote Sensing 60 (2021): 1-10.
> >
> > [2] Chen, Yuxing, and Lorenzo Bruzzone. "Self-supervised Remote Sensing Images Change Detection at Pixel-level." arXiv preprint arXiv:2105.08501 (2021).

---

### Official Review · Reviewer_Mwzz · 2022-07-25
**Useful datasets for studying spatiotemporal change in satellite imagery**

**Rating:** 8
**Confidence:** 4

**Strengths:**

1) Both datasets are indeed unique in the sense that not only pairwise changes but changes of largely varying spatio-temporal scales are available. This allows for a meaningful semantic separation of changes that occur on our planet and is valuable for the development of automated techniques.

2) The description of the datasets in datasheet format, the supplementary material and the webpage is done well, without any unnecessary baggage. The format of the data includes the change events and the full image files of Sentinel-2 along with identifying information that is ready to use and easy to grasp. If I was to develop some technique related to the monitoring of change, I believe this would be an excellent starting point to me.

3) The technique to generate the two datasets is of value and could be readily deployed to generate similar datasets at larger scale.

4) The selected events of road construction and forest fires especially are also of larger relevance (outside the development of change detection technologies) as they relate to important global issues, i.e. climate change.

**Weaknesses:**

1) The authors highlight on several places (for example line 119) how their approach generates change events without any supervision. While it is true that feature learning and grouping of pixels is unsupervised, the assigning of the semantic label to finalize the "change event" is still left to humans.

2) The corresponding labels for the change events are also binary in nature: within each of the two datasets, only a binary decision, e.g. "forest fire? -> yes/no" has been performed and every event labeled "no" is not further described.

3) The pipeline for generating the dataset requires hyperparameter tuning, which the authors acknowledge as a limitation. This might offset the displayed good performance (in terms of quality, but also runtime) somewhat, as multiple trials and tinkering might be required, to achieve a decent result.

4) The fairness in comparison to competing methods is a little questionable (see "Correctness").

**Additional Feedback:**

Nitpicking time: in line 243 there is a "to" missing.
Figure 4 would benefit from a larger size but I acknowledge that NeuRIPS page limitations are an issue here...

Final recommendation: please make a small sample of a few change events from each dataset available for download, so one can get a first impression without downloading several GBs.

**Clarity:**

The paper is well written and easy to understand. The supplementary material is helpful and elaborates on interesting details.

**Correctness:**

Since most (if not all) methods in comparison for the change event representation learning (Section 5.2) are not specifically designed to deal with the change events as formulated by the authors, they are necessarily at a disadvantage. It can not be expected that they can compete if temporal information is simply averaged out over features against a method that makes full use of it.

When it comes to the change detection itself (Section 5.1) the tuning and parameterization of the competing methods is not described so one should be critical about the reported runtimes and (albeit to a lesser extent) also about the reported quality metrics.

While it is clear that the proposed method of the authors can be tuned to be competitive on the OSCD benchmark, it is not given that this performance will generalize to other sources of change/data. Investigating the supplementary material reveals that the authors also compared on the LEVIR-CD dataset, which increases a little the trustworthiness of their results.

More difficult to evaluate is the actual quality/correctness of the detected change events in Calfire and Cairoad, as no human groundtruth of the change segmentation is provided and it seems reasonable to assume that certain change events go undetected or being false positives. In particular puzzling is the statement that 1076 location of fire incidents were reported by Calfire (line 213) while only 204 fire events are positives in the constructed dataset? This is only 20%? I would be curious to learn more about this discrepancy.

From eye-balling some of the change events in the provided data, it seems the segmentations are on a decent enough level to be useful and not necessarily misleading. Given, that the authors also provide the original images along their refined change events, this allows also for evaluations on those quality questions in the future by other independent research groups. If better methods are developed, the revision of the change event portion of the data seems rather straightforward, so I don't consider it a major issue, just something that users of the dataset should be aware of.

**Documentation:**

The documentation in the supplemenatary materials and the dataset itself is good and sufficient. The source-code for generating the datasets has not been made available at the time when I wrote this review, so I cannot judge it. This review is written in good faith that the code will be published eventually and will be easy enough to run.

While I think the hosting on Cornell servers is sufficient for now, I would encourage the authors to mirror their dataset in current and future versions to a free and open dataset repository (for example zenodo.org) which will increase its chances to remain available.

**Ethics:**

The dataset in itself is based on publicly available Sentinel-2 images, which are not at a resolution level that would allow to identify or track individual persons. Furthermore, the authors discourage the use of change detection approaches in general to infringe on privacy. They have also been very transparent with their labeling practices, which involved paying human crowdsource workers.

To me this seems all okay so far and no reason for any further ethical reviewing.

**Relation To Prior Work:**

The paper is set into the appropriate context when it comes to recent work in change detection.

**Summary And Contributions:**

This work introduces two datasets: Calfire and Cairoad that have been refined from about 6 years of publicly available Sentinel-2 RGB imagery. By developing an unsupervised feature learning technique the authors were able to segment regions of change over time, which are highlighted and provided as binary maps between consecutive images. Moreover, using the groundtruth of Calfire, OpenStreetmaps and crowdsourced human annotations, a semantic label has been assigned. The changing pixels together with the semantic label constitute a "change event" which the authors introduce in contrast to the more common pair-wise "change detection". The proposed methods for dataset generation are shown to be efficient and less time-consuming then similar approaches. Lastly, the value of the Calfire and Cairoad datasets as benchmark for the developing of semantic retrieval classification systems is demonstrated.

---

> ### Author Response · Authors · 2022-08-11
> **Review Response**
>
> Thank you for your constructive feedback. We address your concerns in the following response:
>
> ## Weaknesses
> 1. We define a change event to be “a group of pixels over time and space changed by a single event”  L123. The change event does not require the true semantic label to be considered a change event. So our change events are indeed generated without any supervision.
>
> 2. Exhaustive labeling of all the change events would be a very expensive and time consuming  task, but we agree that it is certainly a limitation of our dataset and we have discussed this further in the limitation section. However, researchers interested in other kinds of change events can simply use the change events from our pipeline and label them with their own labels.
>
> 3. While selecting hyperparameters is a limitation, the benchmark is not very sensitive to changes to hyperparameters within a reasonable range (60m-300m $\delta_{st}=3, 30$, 2-8 months $c_t=10, 2.5$). Reducing $\delta_{st}$ to 60m leads to slightly more events 2203 (from 2172). The reason for only a small 1.4% increase is that the region growing is not very sensitive to these hyperparameters. As the majority of components that were connected with  $\delta=20$ still remain collected with $\delta=3$. Additionally experiments on these datasets with small differences from the original dataset do not significantly change performance plots and numbers. We have added this information to the supplementary.
>
> 4. Thank you for your suggestion, hopefully the details we have added (see correctness) answers your concerns.
>
> ## Correctness
>
> **Baselines in sec 5.2**: We agree that averaging over the time dimension is not the best choice. However as shown by the experiments in the supplementary (sec. 8 and fig.8) the **averaging** baseline works better than baselines with temporal information (TST and DTW). There still is a big potential for improvement on representation learning for change events. We posit that better temporal aggregation might be one potential direction of improvement in future work.
>
> **Baselines in sec 5.1**: For the baselines in sec. 5.1, we use the same hyperparameters and inference methods as present in the original implementation of these papers when evaluated on the OSCD benchmark. Additionally, we optimize the data preprocessing step using GPUs for all the baselines resulting in faster preprocessing. So the inference runtime is faster than the original implementations. We have added these details in the supplementary.
>
> **Correctness of detected change events in CalFire**: Due to the low temporal resolution of satellite images, smaller fires out of the 1076 go undetected as the area might look similar within a 1 month window before and after the fire. Similarly due to the low spatial resolution of S2 images, fires with smaller area are not detectable in the images. This is the reason why only 204 larger fire events are detected. Getting false positives is unlikely since we use metadata from the California Fire Department. The metadata looks very reliable as many researchers use it.
>
> **Updating change events**: Yes, the dataset could be evaluated and developed upon in the future by independent researchers. We highly encourage other researchers to use our pipeline to develop and extend the change event datasets with interesting and useful labels.
>
> ## Documentation
> The code is now available and can be found at (https://github.com/utkarshmall13/satellite-change-events).
>
> ## Additional Feedback
>
> **Typos**: We have fixed these 2 issues in the revision.
>
> **Smaller sample datasets**: We have added smaller sample datasets on the dataset page.
>
> New revisions in the paper and supplementary are colored blue during the discussion period.

---

### Official Review · Reviewer_j14f · 2022-07-27
**Review for Change Detection Dataset**

**Rating:** 6
**Confidence:** 4

**Strengths:**

The authors take a compelling approach to a very relevant problem: change detection is crucial in the field of EO, and self-supervision is increasingly being used to distill relevant information within large amounts of imagery.

The authors are obviously very comfortable with implementing novel computer vision algorithms.

The presented dataset contains labels collected via hired enumerators, which would have been an expensive and time-consuming task.

CaiRoad is a great name.

**Weaknesses:**

No code is available for the submission.

The abstract, conclusion, and reported results need to be more quantitative.

There are not enough descriptions for why certain choices were made (see Additional Feedback for more specific comments).

Overall, too much information is packed into too few pages. I understand the page limit is restricting, but the authors need to determine a way to communicate the paper's most crucial information more effectively, while cutting out unnecessary portions.

**Additional Feedback:**

The following comments are presented in rough chronological order.

1. Abstract: What is the dataset size? How many images are contained? What spatial locations are covered? How do benchmarks perform overall? All of this information is crucial to understanding the contributions of the paper, and needs to be specified. This section should be much more quantitative.

2. Introduction: Italicizing and bolding is not necessary in the introduction. I would remove >95% of these font weights, except when you think absolutely necessary.

3. When you described a new self-supervised approach for change detection, is it a completely new approach, or just applied to a new problem here?

4. What type of imagery are you using? Need to state in the abstract or introduction.

5. Lines 87, 89, 91, e.g.: Don't use grouped citations. Split these references up so it's clear what the major contributions are.

6. There are lots of papers on change detection in EO -- An overview of relevant, major papers, and how the field has evolved would be very useful to readers.

7. Lines 103-108: Need more information on what's contained in these other datasets, i.e. the size, spatial extent, imagery, labels, resolution, uses. A table would be effective for this summary.

8. Lines 109-116: See comment above under "relation to prior work. Use references for EO work instead.

Figure 1, sentence 3: Typo/missing word. Caption does not fully describe the different portions of the figure.

Figure 2: Similarly, the caption does not fully describe everything in the figure. Many variables are undefined.

9. Line 120: Sec and all references to tables/figures/etc. must be capitalized. There are many other places where this applied, but I'm only listing the first.

Section 3.1: It seems like this section could be distilled to a single sentence, defining change events as pixel-level reflectance changes. I also don't see how this term materially differs from a common understanding of "change". As a result I don't see how formally defining it is a major contribution of the paper.

3.2: Needs to be condensed, or made substantially more descriptive. What imagery is being used?

10. Line 142: Here, you describe that feature representation of pixels is determined in an unsupervised manner. In sentence 2 of Figure 1's caption, it says that the feature representations are determined in a self-supervised manner.

11. Line 167-168: Why do you need a feature space representation of each pixel instead of a representation for entire image? The same type of contrastive loss can be applied to a full image for unsupervised learning to learn appropriate embeddings. It's possible I'm missing something here, and if so, the case for pixel-level feature representations needs to be made explicit. Explicitly defining feature maps and feature vectors here will help.

12. Line 196: Need to define voxels.

Section 3.2.1: It is unclear what portion of this self-supervised approach is novel. Contrastive losses have been applied in the literature before, as you mention for the SimCLR.

13. Line 188: Is there one threshold for all change detection? Or a separate one for each class?

14. Line 192: How do you prove that you're capturing the correct invariances?

15. Lines 201-2: How are these parameters determined? Need an explanation of how + why.

16. Line 208: How much S2 imagery is used? Need a description on the frequency/extent/etc. Also, why do you not use the near-infrared band? This is the band most common used in burned area detection approaches.

17. Line 222: Similarly, why do you not use the shortwave infrared bands? These bands will be particularly useful for road detection.

18. Lines 238-241: How is road change defined to enumerators? As state previously, change may be partial and may extend over multiple timesteps.

Figure 3: There is not enough information in the Figure caption to understand what's being presented. I see the note about more information being contained in the Supplement, but you can't introduce a figure in the main text and not describe it in the Supplement.

19. Line 261: Why is this for fairness? Explain.

20. Line 255: OSCD is used for unsupervised learning and benchmarking. Why is all train + test data used for training? What is performance measured over then?

21. Line 268: Need to explain why you're using an estimate here and how the estimate was calculated if you're going to use it as one of your main reported findings.

22. Line 297: Need more explanation of how features are averaged across time. Won't this flatten the necessary time dimension for change detection?

Conclusion: Like the abstract and introduction, needs to be more quantitative for both the dataset + benchmarks.

**Clarity:**

The writing quality of this paper is okay. There are no major issues, but there is lots of repeated information and sentence structures.

**Correctness:**

As far as I can tell, everything in this submission is correct. The dataset is constructed in a sound way, and the models seem to be implemented correctly.

**Documentation:**

Yes, there is sufficient data on data collection.

Unfortunately, the authors did not publish any of their code for curating the dataset or implementing their change detection models.

**Ethics:**

N/A.

**Relation To Prior Work:**

This is a portion of the paper that can be improved. Self-supervision is being adopted for more and more EO tasks, but the authors do not discuss recent efforts to apply these computer vision-developed approaches to the realm of remote sensing. All discussion of self-supervision methods + papers pertain to non-EO tasks.

**Summary And Contributions:**

Overall. this paper presents interesting work + a dataset, but it is not described in a way that gives the reader a clear view of (1) what's being presented in terms of the dataset, 2) what portions of the methodology are novel, and (3) why certain decisions were made by the authors.

Furthermore, the abstract and the conclusion need to be reworked, and the reporting of findings needs to be more quantitative.

With additional work, this will be a nice submission for a future conference.

---

> ### Author Response · Authors · 2022-08-11
> **Review Response**
>
> Thank you for your detailed and constructive  feedback. We address your concerns in the following response:
>
>
> ## Weaknesses
> **Code**: The code is now available and can be found at (https://github.com/utkarshmall13/satellite-change-events).
>
> **Quantitative information**: Thank you for the suggestion. We have added more details in the abstract, conclusion and results. (see additional feedback)
>
> **More description**:  Thank you for all your suggestions. We have edited the revision based on all your suggestions (see additional feedback).
>
> **Cutting out unnecessary portions**: We have made changes based on your suggestions. We have also tried to cut some redundant portions. Please let us know if the revision reads better to you.
>
> ## Relation To Prior Work:
>
> **Discussion about self-supervision in EO**: We have added a new discussion of self-supervised EO papers in Sec. 2. If you have other papers in mind that we should discuss, please let us know. We will be happy to add them to the revision.
>
> ## Additional Feedback
>
> 1. We have added the following information to the revised abstract. “The CaiRoad dataset has a total of 28015 change events with 2259 road construction events from the city of Cairo and the CalFire dataset has 2172 change events with 204 labeled fire events in California. These new datasets are challenging for many existing methods. For example the best performing model has a retrieval precision@25 of 0.46 on CaiRoad.”
> 2. Thank you for the suggestion. We have incorporated your suggestions.
> 3. SimCLR is a known approach for self-supervised learning. Our contribution is not SimCLR itself, but a new architecture and the way that SimCLR is used at individual pixels efficiently using equivariant geometric transforms.
> 4. We have updated the abstract and intro with this information. We are using Sentinel-2 satellite imagery (with 10m spatial resolution and 1 month temporal resolution).
> 5. Thank you for your comment. We have made this change to the paper.
> 6. We have added some more change detection papers in the related work section and made the subsection more clear. It would also be helpful to know about any important papers that you think we have missed. We would be happy to add them.
> 7. We have added a table with this information in the supplementary (table 1). However, Since our dataset is not a human labeled binary change dataset, it will not be directly comparable to these datasets. We have made this clear in the revision.
> - **Figure 1 and 2**: Thank you for your comment. We have updated the figure captions with detailed information.
> 8. We have added a new discussion of self-supervised EO papers in Sec. 2. If there are others we have missed, please let us know. We will be happy to add them to the revision.
> 9. Thanks for your comment. We have made this change in the paper.
> - **Section 3.1** While a change mask can be defined as a “pixel-level reflectance change”, a change event is  not just that. Change events can span multiple time-steps and a single change event happens  due to a single real-world event (e.g., fire, road construction). Thus, “pixel-level reflectance changes” are not sufficient to describe it. To summarize, change events differ from change masks in two ways, (1) change events can span multiple timesteps, (2) unlike a change mask, a single change event is due to a single semantic event and thus is a more useful structure. The semantic meaning of our change events is why we believe the dataset we propose makes a useful contribution.
> - **Section 3.2** Thank you for your suggestion; we have added more information in this section.
> 10. The two terms are not contradictory since self-supervised learning is a type of unsupervised learning, where the unlabelled data itself is used to train the networks with supervised losses..
> 11. Since change detection is a per-pixel classification, we need a feature representation for each pixel. Even within a single image the pixel representation can be drastically different as they could be representing different characteristics. Thus, having a representation for images instead of pixels would lead to very coarse change detection. We have made this explicit in the revision.
> 12. Thank you for the suggestion, we have made this clear in revision. A Voxel is a pixel in 3 dimensions.
> - **Novel contribution**: The loss function itself (SimCLR) is not a novel contribution of the paper. But the architecture for change detection and the way that SimCLR is used at individual pixels, efficiently using equivariant geometric transforms, is a novel contribution of this work.
> 13. Yes, there is a single threshold for all classes of change detection. As stated later (l261), we use Otsu’s thresholding to find this threshold.
>
> (continued in next comment.)

---

> > ### Author Response · Authors · 2022-08-11
> > **Review response (cont.)**
> >
> > ## Additional Feedback
> > 14. Our method's ability to capture the correct invariances is the reason why it works better than the baselines. In the paper we show this implicitly by looking at the change detection performance. However, we plan to do an experiment showing invariances to the desired augmentations such as photometric transformations, and equivariance to others such as rotation, translation and scaling. We will update the revision and notify you as soon as we get the results.
> > 15. The hyperparameters values were selected by qualitatively inspecting created events on a very small subset of input. We have added this information to the revision. The benchmark is not very sensitive to changes to hyperparameters (60m-300m $\delta_{st}=3, 30$, 2-8 months $c_t=10, 2.5$). Reducing $\delta_{st}$ to 60m leads to slightly more events 2203 (from 2172). The reason for only a small 1.4% increase is that the region growing is not very sensitive to these hyperparameters. As the majority of components that were connected with  $\delta=20$ still remain collected with $\delta=3$. Additionally experiments on these datasets with small differences from the original dataset do not significantly change performance plots and numbers. We have added this information to the supplementary.
> > 16. Thank you for the suggestion we have added the following information in revision. For CalFire we used a total of 25688 S2 images and 24830 S2 images? for CaiRoad. We use the R (664.5nm), G (560nm), and B (496.6nm) bands from Sentinel-2. The information about extent of the dataset can be found in the metadata of the dataset. More statistics about the dataset is present in sec. 4. of the supplementary.
> > 17. We agree that using NIR or SWIR band could have been useful for road and fire detection. However, there are several advantages to using RGB band. Firstly, using the RGB band allows for the use of pretrained networks from computer vision since these tools are developed on RGB imagery. Secondly, it should be noted that the CaiRoad benchmark was developed using human annotators who looked at RGB images. Using SWIR and NIR bands for annotations would require remote sensing experts that can interpret these bands and would result in more expensive annotation. Finally, our dataset does not limit other downstream works from using our annotations in conjunction with more bands.
> > 18. The annotators are explained that construction changes can extend over multiple timesteps and are shown various examples in instructions and tutorials (Fig. 5 and 6 supp.), resulting in the high quality of desirable data. We have added this clarification in the revision.
> > - **Figure 3**: In figure 3, the events are represented in $\langle V_{1\cdots l}, C_{1\cdots l-1} \rangle$ format. $V[0]\cdots V[3]$ are RGB satellite images providing visual information and $C[0]\cdots C[2]$ are binary change masks showing changes between consecutive satellite images (white is change, black is no change). For example, $C[1]$ shows the change between $V[1]$ and $V[2]$. We have added this in the revision.
> >
> > 19. Since the range of difference values can vary across methods, each method needs a different optimal threshold. Therefore for fairness, we use Otsu’s thresholding for all the methods. The threshold maximizes interclass variance for two classes above and below the threshold. We have added this discussion in the paper
> >
> > 20. Since all the methods are based on unsupervised learning (**no change masks are used during training**), the same dataset can be used to learn feature representation and evaluation. So we use the full dataset for unsupervised training and **also testing**. This is also similar to how other unsupervised prior works use this dataset [40].
> >
> > 21. We used an estimate to not waste the compute by running the baseline on our full dataset which would take 14 days. We evaluated performance on a smaller subset of 60 temporal pairs where on average it took 51 seconds for change detection. In total there are approximately 24000 such pairs in CaiRoad. So the estimate comes to 14 days. We have added this information in the paper.
> >
> > 22. Averaging the feature across time will flatten the time dimension. However, since the change pairwise masks are explicitly provided to the network, that information is not being reduced. We have added this to the discussion in sec 5.2
> >
> > - **Conclusion**: Thank you. We have added more information to the conclusion
> >
> > New revisions in the paper and supplementary are colored blue during the discussion period.

---

### Official Review · Reviewer_tPUf · 2022-07-28
**Good data set, paper needs work**

**Rating:** 7
**Confidence:** 4

**Strengths:**

The spatio-temporal dataset presented by the authors are novel, and has value in itself especially for researches working on object detection, anomaly detection, change detection, online learning, and remote sensing, among others. While I have issues with the writeup, I see value in the dataset given the lack of spatio-temporal datasets (see review section: discussion on previous work)

**Weaknesses:**

The fatal weakness to the paper is the description of the baselines in particular the "the baseline "Pretrained imagenet"".Without the architecture of the pretrained network (I am assuming that it is a neural network) that description is uselss. Also the note that "No Training involved" appears impossible; ImageNet is a 1000-class problem, and without fine tuning what exactly is the "pretrained network" trying to discriminate?

We need more clarification regarding this.

Also, the large difference between the recall and F1-scores seem to indicate that that the baseline methods have very poor precision and produces a large number of false positives. Can the authors provide more insights on the false positives and if they all fall into some natural grouping (seasonal change, shadows in the picture, occlusion, etc?)

I would also have liked to see more detail on the self-supervised change detection network, namely some example images of that were considered "similar"

**Additional Feedback:**

While I like the dataset, and I do see the value the dataset would add to the machine learning community, the write-up in particular the benchmarking process requires more work in particular the description of the baselines. While I am inclined to accept the submission on the merits of the value of the dataset, I do think that more work is needed in the writeup as noted above

**Clarity:**

Unfortunately, as noted in the previous sections, there are a number of details that are unclear in the paper, namely the imagenet Baseline method, the cloud processing, some discussions on the false positives,

**Correctness:**

The dataset appears to be constructed in a sound way, however I am concerned about the high number of false positives reported in the baselines. Without a breakdown of the nature of the false positives, it is hard to tell if there is a systemic problem in the dataset, or the false positives comes from the weakness in the baseline methods and the dataset reveals areas where improvements can be made to the baseline algorithms.

I have issues with the benchmarks, elaborated in the previous section.

**Documentation:**

The authors might want to note the physical characteristics of the dataset, in particular the spatial resolution of the RGB camera of Sentinel-2 (10m) so that users of the dataset have a realistic expectation of the amount of change needed before it results in changes in the image.
Also I note in the supplementary material, the authors mention that 80% of the changes are caused by clouds and are filtered out? How is the filtering done? Are the authors using Copernicus' IDEPIX to detect clouds and filtering them that way? The authors noted that they use representational learning so that the change detector is robust to illumination changes; It would be helpful if the authors provide examplars of the illumination changes and report the performance of the change detector.

**Ethics:**

I do not have any concerns regarding the ethics of the dataset.

**Relation To Prior Work:**

The authors did not reference other spatio-temporal datasets. To be fair to the authors, to the best of my knowledge there are not many publicly available spatio-temporal datasets. Off the top of my head, the only ones i know of are Pheneo4D (temporal 3d scans of maize growing, https://www.ipb.uni-bonn.de/data/pheno4d/ , Soil-moisture of the Tibetan Plateau https://doi.org/10.6084/m9.figshare.7996448, and Global Spatiotemporal data for 2019-Novel Coronavirus Covid-19 Cases and deaths, https://data.humdata.org/dataset/2019-novel-coronavirus-cases, all of which are fundamentally different from the dataset presented.

A more related dataset would be the Borneo forest logging dataset, however, to the best of my knowledge it is not publicly available to researchers. https://journals.plos.org/plosone/article?id=10.1371/journal.pone.0101654




**Summary And Contributions:**

The authors are presenting 2 spatio-temporal datasets taken using remote sensing satelites, The authors describe the methods for annotating change events, and a pipeline the authors used to automate the annotation of the dataset.

---

> ### Author Response · Authors · 2022-08-16
> **Review response**
>
> Thank you for your detailed and constructive feedback. Please see our responses to your concerns below.
>
> ## Weaknesses
>
> **Description of baselines**: We use a ResNet-18 for all the SimCLR and Pretrained ImageNet baseline (please refer to sec. 3.3 in supplementary for more information). We have also added the architecture information in the main paper so that the descriptions of baselines becomes more useful.
>
> We remove the final classifier layer and use the penultimate layer features from a pre-trained imagenet resnet-18 as a generic feature extractor in many cases (such as zero-shot learning). Even without fine-tuning, the representation has some discriminative power and can be used for retrieval tasks. The results in figure 4 show that in many cases, it is a strong baseline.
>
> **F1 scores**: The reason for low F1 score is that the true change labels are not exhaustive and only urban changes are annotated in OSCD. Thus, the false positives involve other kinds of changes, like natural disasters or seasonal changes. Our method’s high recall shows that it recovers a big fraction of the annotated changes while having a higher F1 score than the baselines. We’ve added some examples of such change masks in the revised supplementary (see sec. 5 (Justification for False Positives.)). It shows that many such “false positives” are not because of our method but because OSCD does not have labels for non-urban changes.
>
> **Examples of similar images**: In contrastive learning, the key idea is to pull together representations of two different augmentations of the same image and push apart augmentations of different images (or instance classification). So images that would be considered similar would be two different augmentation (photometric in our case). One such example is shown in Fig. 2, the two images labeled $p^0(I)$ and $p^1(I)$.
>
> ## Correctness
>
> **Breakdown of False positives**: A high number of false positives by the baselines on our benchmark are occurring not because of a problem in the benchmark, but because of the challenging nature of the dataset. This is why the conventional computer vision baselines are not very good at this problem. Learning the representation for change events is challenging for two reasons: (1) the temporal direction of change is hard to model with the baselines, (2) changes of the same categories have very different shapes and sizes. (3) change events of other types might overlap with regions with roads.
>
> We have added in illustration with retrieved examples in the supplementary (section 7) that shows that the dataset is indeed challenging and existing baselines are not very strong. These false positives either overlap with a road construction or a road (no construction). Sometimes the false positives change events near a road construction and their features get influenced by the construction near it due to the receptive field.
>
>
> ## Clarity
>
> We have updated the writeup with more information about the baseline and the benchmark based on your suggestions. We also are continuing to edit the revisions based on feedback.
>
> ## Relation To Prior Work
>
> Thanks for the suggestion; we believe the datasets for soil moisture and forest logging are the most relevant and we have added them to the discussion.
>
> ## Documentation
>
> **More information to the users of the dataset**: Thanks for your suggestion, we have added this information in the datasheet in sec. 2.1 of the supplementary.
>
> **Cloud detection**: We use the QA bands available with the sentinel-2 SR dataset to filter out the cloud. The band is named “S2_CLOUD_PROBABILITY” on EarthEngine. It stores a probability value  of presence or absence of  cloud, and we use 0.5 as the threshold for masking. We have added this to the details about cloud removal in the supplementary.
>
> **Example of illumination change**: We have added examples of illumination changes and we show that our method is indeed robust to such changes. Please refer to sec. 5 (Qualitative Evaluation of Change Detection) for these examples. We show that our method is robust to illumination changes due to shadows and time of year. It is also able to ignore illumination changes while detecting real changes.
>
> ## Additional Feedback
>
> Thank you for your useful suggestions and feedback. We have incorporated the information and the writing/explanation suggestions you and other reviewers have requested to improve the presentation.
>
> New revisions in the paper and supplementary are colored blue during the discussion period.

---

### Meta-Review · Area_Chair_wFoE · 2022-09-12

**Recommendation:** Accept
**Confidence:** 5

**Metareview:**

This work presents a novel resource for detecting changes in aerial imagery. The paper is well written and contains significant amount of experiments and baselines for demonstrating its usefulness. The paper considerably improved upon reviewers feedback who are unanimously positive about this work.

---

### Decision · Program_Chairs · 2022-09-16

Accept